

# Circular RNAs as potential biomarkers and therapeutics for cardiovascular disease

Weitie Wang, Yong Wang, Hulin Piao, Bo Li, Maoxun Huang, Zhicheng Zhu, Dan Li, Tiance Wang, Rihao Xu and Kexiang Liu

Department of Cardiovascular Surgery, The Second Hospital of Jilin University, Jilin, China

## ABSTRACT

Circular RNAs (circRNAs) are genetic regulators that were earlier considered as "junk". In contrast to linear RNAs, they have covalently linked ends with no polyadenylated tails. CircRNAs can act as RNA-binding proteins, sequestering agents, transcriptional regulators, as well as microRNA sponges. In addition, it is reported that some selected circRNAs are transformed into functional proteins. These RNA molecules always circularize through covalent bonds, and their presence has been demonstrated across species. They are usually abundant and stable as well as evolutionarily conserved in tissues (liver, lung, stomach), saliva, exosomes, and blood. Therefore, they have been proposed as the "next big thing" in molecular biomarkers for several diseases, particularly in cancer. Recently, circRNAs have been investigated in cardiovascular diseases (CVD) and reported to play important roles in heart failure, coronary artery disease, and myocardial infarction. Here, we review the recent literature and discuss the impact and the diagnostic and prognostic values of circRNAs in CVD.

**Subject** Bioinformatics
**Keywords** Circular RNAs, Biomarkers, Cardiovascular disease

# BASIC REPORTING
## Introduction

Almost 70% of the human genome is transcribed. However, protein-coding genes account for only 1–2% of the human genome, whereas the majority of transcripts are noncoding RNAs (ncRNAs) (*Djebali et al., 2012*). Noncoding RNA can be divided into two types: housekeeping ncRNAs and regulatory ncRNAs. The latter consists of microRNAs (miRNAs, 8.13%), small nuclear RNAs (snRNAs, 8.21%), small interfering RNA (siRNAs, 4.07%), and long noncoding RNAs (lncRNAs, 69.42%) (*Frankish et al., 2018*) (Fig. 1). Noncoding RNAs can also be categorized into two families based on size: large (>200 nt) or small (<200 nt) (*Taft et al., 2010*). Between the two families, the large ncRNAs are produced by the same transcription and splicing machinery as mRNA, whereas small RNAs are always transcribed from genomic loci and processed by specific nucleases. Circular RNAs (circRNAs) are a type of ncRNA that are abundant and stably exist in living organisms. They are different from linear RNAs because of their covalently closed-loop structure, which makes them gene expression regulators, and have potential clinical therapeutic values (*Fan et al., 2017*).

Corresponding author
Kexiang Liu, kxliu64@hotmail.com

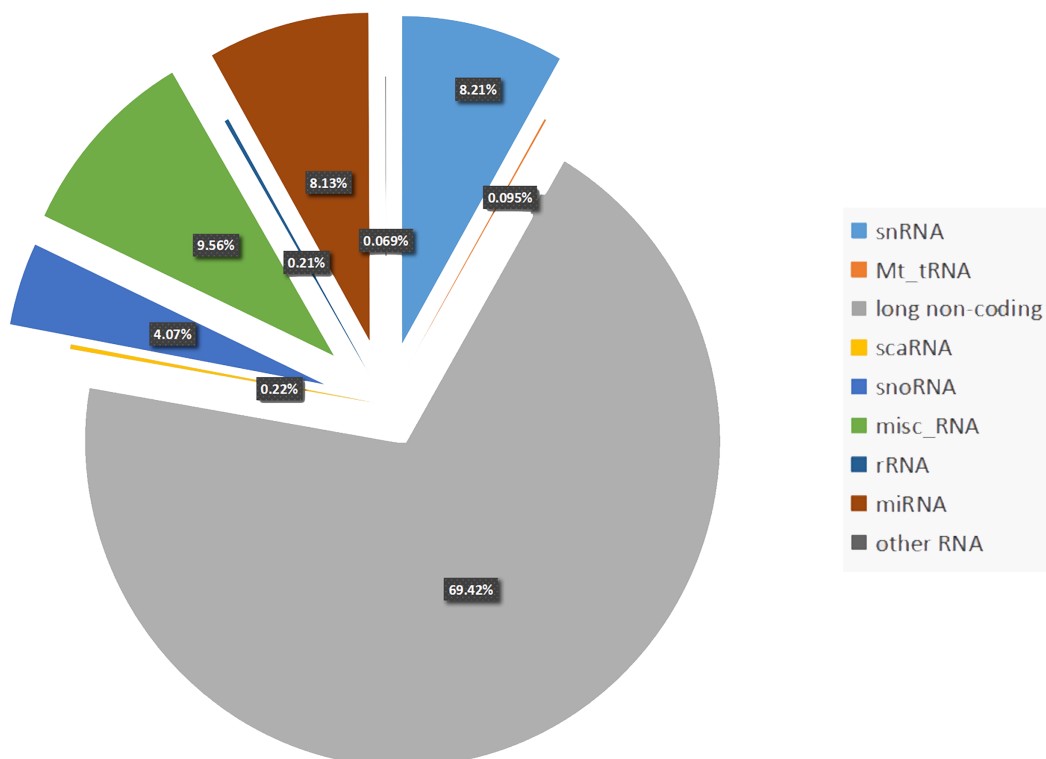

**Figure 1 Percentage of each non-coding RNAs class.** Percentage of the regulatory non-coding RNAs class. Among these non-coding RNAs, long non-coding RNAs occupy the largest proportion (69.42%). The following proportion are snRNAs (8.21%), Mt_tRNAs (0.095%), scaRNAs (0.21%), snoRNAs (4.07%), misc_RNAs (9.56%), rRNAs (0.22%), miRNAs (8.13%), other RNAs including (sRNAs, Mt_rRNAs, ribozyme and vault RNAs) account 0.069%.

With the discovery that ncRNAs are necessary for heart physiology, the important roles of these molecules in cardiovascular disease (CVD) has become evident (*Barwari, Joshi & Mayr, 2016*). The main groups of ncRNAs, including miRNAs, lncRNAs, and circRNAs have been reported to be relevant for cardiovascular physiology and disease (*Fan et al., 2017*; *Greco, Gorospe & Martelli, 2015*; *Wojciechowska, Braniewska & Kozar-Kamińska, 2017*). Particularly, in the past several years, miRNAs and lncRNAs were shown to be critical contributors to cardiovascular pathophysiology. Moreover, they were reported to be potential biomarkers for several diseases, as they were abundant and stable as well as evolutionarily conserved in tissue (liver, lung, stomach), saliva, exosomes, and blood (*Wang, Chen & Sen, 2016*; *Yang et al., 2017b*; *Shi & Yang, 2016*; *Li et al., 2018a*).

Cardiovascular diseases are a major cause of death worldwide. Early diagnosis and treatment can reduce major adverse cardiovascular events (*Kim et al., 2011*). Thus, efficient biomarkers are important for diagnosis of CVD especially in cardiac injury. Cardiac troponins T/I (cTnT/I) (*Heil & Tang, 2015*) and creatine kinase myocardial band isoenzyme (CK-MB) have been widely used as biomarkers for myocardial infarction (MI), while brain natriuretic peptide (BNP) has been used for heart failure (HF) for a long time (*Calzetta et al., 2016*). Ideal biomarkers must be sensitive and specific for specific diseases and should not be influenced by heterogeneity of the heart-associated disease,

lifestyle, patient's age, and genetic background (*Kondkar & Abu-Amero, 2015*; *Enroth et al., 2014*). Currently, highly sensitive biomarkers are urgently needed for earlier diagnosis of CVDs.

Cardiac physiology is a complex balance between electrical stimuli and chemicals related to the mechanism of contraction. Recently, ncRNAs were reported to be ideal regulators of the cardiovascular system, and circRNAs were linked to CVD (*Thum & Condorelli, 2015*; *Uchida & Dimmeler, 2015*; *Vegter et al., 2016*). CircRNAs are a class of ncRNAs that are more stable than linear RNAs (*Suzuki & Tsukahara, 2014*), as they form a covalently (*Wilusz & Sharp, 2013*) closed-continuous loop, which is resistant to RNase R activity (*Gao, Wang & Zhao, 2015*). They were thought to be aberrant RNA splicing products with no functions (*Sanger et al., 1976*) for nearly two decades until bioinformatics methods and RNA sequencing (RNA-Seq) technologies indicated their association with specific biological processes (*Salzman et al., 2012*) These circRNAs are always stored in the cytoplasm (*Jeck et al., 2013*) and only a small part exists in the nucleus (*Ebbesen, Hansen & Kjems, 2017*). Many studies have found that they participate in cell proliferation (*Zhao et al., 2015*), migration, and invasion (*Zhong, Lv & Chen, 2016*; *Ng et al., 2016*; *Wang et al., 2017b*; *Xu et al., 2017*), gene transcription (*Memczak et al., 2013*), and function as gene expression regulators, miRNA sponges (*Memczak et al., 2013*; *Hansen et al., 2013*), and RNA-binding protein (RBP) sponges (*Greene et al., 2017*; *You et al., 2015*). CircRNAs are highly stable and detectable in body fluids (*Zhang et al., 2018a*) as well as abundantly expressed (*Wang et al., 2014*; *Zhang et al., 2018a*) and evolutionarily conserved (*Noto, Schmidt & Matera, 2017*) in humans, making them better biomarkers than linear RNAs with potential value in clinical diagnosis, therapeutic, and prognostic applications (*Wang et al., 2018*; *Kun-Peng et al., 2018*; *Xing et al., 2018*).

Recent studies have reported the potential of circRNAs as biomarkers (*Kun-Peng et al., 2018*; *Xing et al., 2018*; *Haque & Harries, 2017*; *Zhang et al., 2017*) to detect diseases in *Drosophila* and humans, especially cancer. In this review, we discuss the characteristics and functions of circRNAs. Moreover, we review the current research on circRNAs in CVDs, providing evidence for the significance of circRNAs in CVD diagnosis and clinical treatment.

## STUDY DESIGN

### Survey methodology

Scholarly articles that were reviewed in this paper were searched in journal databases and subject-specific professional websites. The search terms that were used to search the articles included circRNAs, potential biomarkers, therapeutic, and CVD. Inclusion criteria for selected articles required that articles be directly related to the topic on circRNAs and be peer reviewed.

## VALIDITY OF THE FINDINGS

### Biogenesis of circRNAs

In contrast to linear RNAs, circRNAs lack a 5′ cap and 3′ polyadenylated tail. They are spliced from pre-mRNA and specially formed with closed-covalent bonds

(*Wilusz & Sharp, 2013*). There are three different types of circRNAs: exonic, intronic, and exon–intron circRNAs (*Li et al., 2015a*). Recent evidence indicated that exonic circRNAs are formed when a 3′ splice donor attaches to the 5′ splice acceptor of a single exon. This type of circRNA accounts for more than 80% of all types of circRNAs. Occasionally, the start of an upstream exon splices and attaches to the other end of a downstream exon, a process referred to as back-splicing process. Thus, a spliced donor joins an acceptor site to form a circular transcript, after which introns are spliced out (*Zhang et al., 2016*) (Fig. 2A). However, if the intron is retained during this process, an exon–intron circRNA is formed (Fig. 2B). The intronic circRNAs always form from intron lariats, which contain a single unique 2′–5′ linkage. GU-rich sequences near the 5′ splice site and C-rich sequences near the branch point bind into a circle, and then exonic and intronic sequences are cut out with the remaining introns being brought together to form intronic circRNA (*Taborda, Ramírez & Bernal, 2017*) (Fig. 2C).

Emerging studies report another mode of circRNA biogenesis that depends on RBPs such as Quaking (QKI) (*Conn et al., 2015*) and Muscleblind (MBL) (*Ashwal-Fluss et al., 2014*), which bridge two flanking introns close together (Fig. 2D). Another RBP, an adenosine deaminase acting on RNA-1 (ADAR1) (*Song et al., 2016*), can prevent circRNA formation by melting the stem structure (Fig. 2B). Meanwhile, some studies have revealed that heterogeneous nuclear ribonucleoprotein (hnRNP) and serine-arginine (SR) proteins regulate the expression of circRNAs in *Drosophila* (*Kramer et al., 2015*), suggesting that RBPs also play an important role in regulating the levels of circRNAs.

## CircRNAs as miRNA sponges

CircRNAs have been reported to function as miRNA sponges by mediating the downregulation or upregulation of miRNA target gene expression (*Thomson & Dinger, 2016*). They are found to negatively regulate gene expression by absorption and sequestration of miRNA molecules (Fig. 3). MicroRNAs are a class of small ncRNAs that play an important role in regulating gene expression through the repression of mRNAs. ciRS-7/CDR1as and Sry are representative circRNAs that function as miRNA sponges (*Huang et al., 2018b*; *Li et al., 2018b*). They have 16 miR-138 and 74 miR-7 binding sites, respectively. Overexession or knockdown of ciRS-7/CDR1as or Sry results in synchronous increased or decreased expression of the relevant miRNAs, respectively. It is interesting that some circRNAs contain multiple binding sites for a single miRNA. For example, circHIPK3 contains nine binding sites for growth-suppressive miRNAs (*Zheng et al., 2016*). On the other hand, some circRNAs, unlike circHIPK3, contain only two binding sites for miR-124, and yet are able to regulate the function/expression of this miRNA. Thus, although multiple binding sites may not be a prerequisite for their regulatory function, circRNAs with multiple binding sites may affect the expression of more miRNA targets. It is still unclear whether a single miRNA binding site is sufficient for the miRNA sponging function of circRNAs.

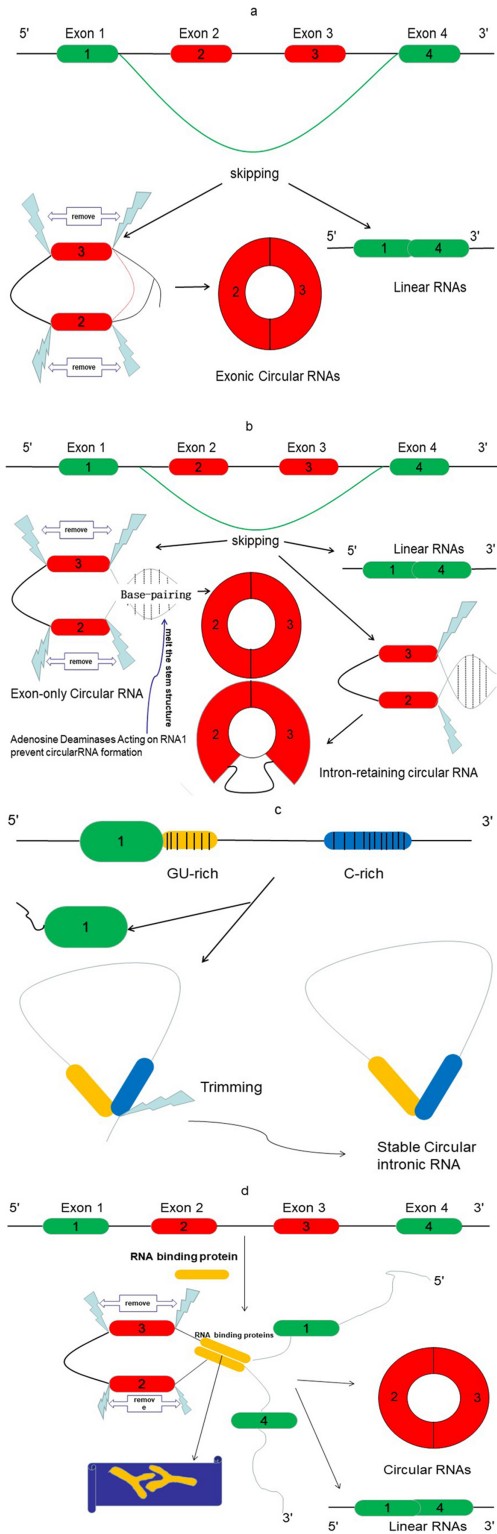

**Figure 2 The proposed models of circRNA formation.** (A) Direct lariat-driven cyclization. Exon splicing generates a lariat structure. The 3′ splice donor site of exon 1 covalently links to the 5′ splice acceptor of exon 4. Circular exonic RNA is formed after removal of intronic sequence. (B) Intron-pairing-driven circularization. Direct base-pairing of the introns flanking inverted repeats or ALU

**Figure 2** (continued)
elements leads to the formation of a circular structure. The introns are removed or retained to form exonic circRNA or exon-intron circRNA. (C) The circular intronic RNAs are generated from lariat introns that can escape debranching; 7 nt GU-rich sequences near exon 1 (yellow box) and 11 nt C-rich sequences near exon 2 (blue box) form the circular intronic RNAs, by avoiding debranching and become a stable circRNA. (D) RNA binding protein (RBP)-driven circularization: circRNA is formed through RBPs (Y-shape), and introns are removed.

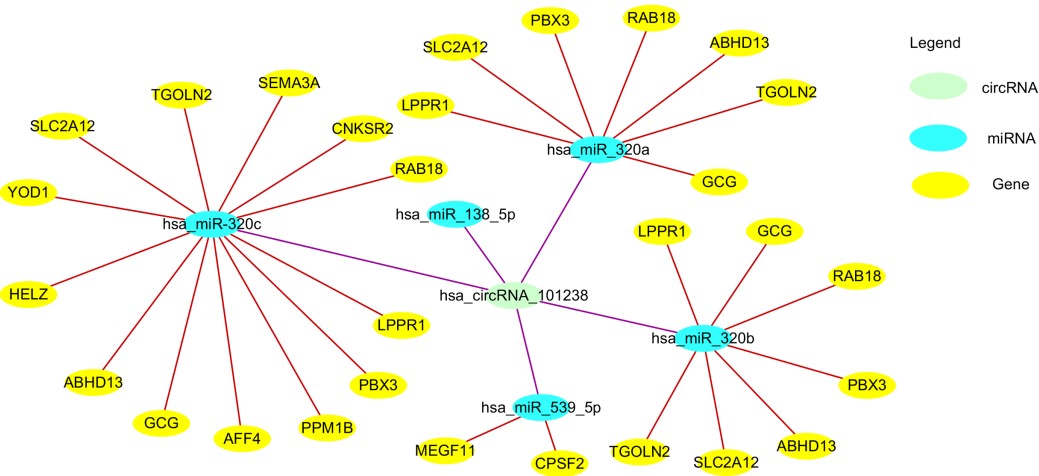

**Figure 3 The circ101238/miR/target gene regulatory network.** circ101238 negatively regulate gene expression by absorption and sequestration of the five main miRNA molecules.

## CircRNAs as RBP sponges

RNA-binding proteins play an important role in post-transcriptional regulatory processes associated with cell differentiation, proliferation, apoptosis, and oxidative stress (*Abdelmohsen et al., 2008*). Study shows that specific RBPs can influence the lifecycle of an mRNA. Recent research showed that circRNAs could function as "super-sponges" with a given RBP and thereby regulate the target gene (*Dudekula et al., 2016*). The hsa_circ_0000020 contains six binding sites for human antigen R (HuR) and 10 sites for fragile mental retardation protein (FMRP), while the hsa_circ_0024707 contains 85 binding sites for argonaute 2 (AGO2). These circRNAs can store, sort, deliver, and regulate RBPs owing to the high density of binding sites (*Hentze & Preiss, 2012*). Another study showed that circPABPN1 can competitively bind to HuR to prevent polyadenylate-binding protein 1 (PABPN1) mRNA from binding to HuR in order to influence the translation process (*Abdelmohsen et al., 2017*). Another recent study suggested that circular antisense non-coding RNA in the INK4 locus (ANRIL) transcripts may regulate the INK4/ARF coding transcripts by competitive splicing (*Burd et al., 2010*).

## CircRNAs as transcriptional regulators

Misregulation of alternative splicing is associated with the aberrant expression of splicing factors. Research shows that circRNAs can act as splicing isoforms and function in

regulating alternative splicing. Nuclear exon–intron circRNAs (EIciRNAs) can regulate transcription. The intronic sequence of these specific circRNAs can interact with the U1 component of the spliceosomal machinery and promote the expression of the target genes through recruiting RNA polymerase (Pol) II to the promoter region of genes (*Qin, Wei & Sun, 2018*). For example, the circRNA ciankrd52 can act as a positive regulator of Pol II to promote ankyrin repeat domain 52 (ANKRD52) gene transcription (*Zhang et al., 2013*). CircRNAs can also influence the translation of the cognate linear mRNA and thereby regulate protein expression. For example, circPABPN1 can interact with its cognate mRNA, PABPN1, to reduce the PABPN1 translational efficiency (*Abdelmohsen et al., 2017*). The circRNA EIciEIF3j could combine with snRNPs (small nuclear ribonucleoproteins) and Pol II to promote its parent gene EIF3J transcription (*ElAntak et al., 2010*; *ElAntak et al., 2007*). Recent reports have shown that ci-mcm5 and ci-sirt7 can enhance the expression of RNA associated with tumorigenesis (*Li et al., 2015a*). In addition, circRNAs can inhibit mRNA maturation. For example, circANRIL can bind to the C-terminal lysine-rich domain of pescadillo (PES)1, thereby inhibiting pre rRNA processing and impairing ribosome assembly and translation processes (*Burd et al., 2010*).

## CircRNAs as competitors of linear splicing

A common pre-mRNA may produce many isoforms. The MBL gene contains sequences that form a circRNA transcript having binding sites for MBL itself. Subsequently, in an autoregulatory manner, circMBL influences the selective splicing of the MBL mRNA. Further, MBL can interact with circMBL and its flanking introns to promote exon circularization. In addition, the competition between canonical splicing and circRNA generation is evident from the concomitant decrease in circRNA and increase in linear splicing (*Ashwal-Fluss et al., 2014*).

## CircRNAs as protein/peptide translators

Although classified as ncRNA, a recent report showed that circRNAs could be translated into proteins. In fact, the human transcriptome seems to contain many circRNAs with coding potential (*Yang et al., 2017a*). These have been associated in vivo with the translation of polysomes of *Drosophila*. Circ-ZNF609 was able to translate a GFP protein during myogenesis because it contains an open reading frame at the start codon as the linear transcript (*Legnini et al., 2017*). However, because these target circRNAs have no free 5′ and 3′ ends, this translation process occurred through rolling circle amplification (RCA) mechanism, driven by internal ribosome entry sites (IRES) and in a 5′-cap-independent manner. The circ-FBXW7 was found to encode functional proteins in human U251 and U373 cell lines (*Yang et al., 2018*). eIF4G2, YTHDF3, and N6-methyl adenosine residues, as well as methyltransferase METTL3/14 were found to promote the translation initiation of circRNAs (*Yang et al., 2017a*). Although the protein translation efficiency of target circRNAs is lower in human and murine cells, the hepatitis D virus antigen (HDAg) encoded by circRNAs exists after infecting eukaryotic cells. In addition, mRNA modifications of m6A, Ψ, and m5C are important for circRNA translation process, and can alter the efficiency and fidelity of translation (*Gilbert, Bell & Schaening, 2016*).

A recent report has shown that circRNAs can also promote protein-protein interactions. For example, CDK2 is a key marker of cell progression from G1 to S phase (*Du et al., 2017*). Circ-Foxo3 can interact with CDK2 via p21 to inhibit cell cycle progression. Recently, a circRNA was shown to decrease the nuclear translocation of its co-localized proteins, ID1 and E2F1, and to also decrease the distribution of HIF1α (*Corcoran & O'Neill, 2016*), indicating that circRNAs can also affect protein localization.

## CircRNAs as potential diagnostic and prognostic biomarkers of CVD

Linear RNA molecules have been reported as potential biomarkers in several diseases, especially cancer (*Wu et al., 2017*). However, circRNAs were identified to have more advantages than linear RNAs as biomarkers (*Xia et al., 2018*). It is an ancient, evolutionarily conserved feature in humans and mice, as well as *Drosophila*. They are more abundant than expected and more stable than linear RNAs due to the covalently closed-loop structures, which can resist RNA exonuclease and RNase R activity. In addition, the half-life of stable ncRNAs is about 48 h, which is much more than that of mRNAs (10 h) (*Ayupe & Reis, 2017*). Moreover, they are located in the cytoplasm, which can be easily acquired and examined.

For a long time, biomarkers have been used for early diagnosis and as indicators of the severity of abnormal processes as well as to predict treatment outcomes. An ideal biomarker must be sufficiently variable under normal and diseased conditions for efficient diagnosis. At the same time, it must be easily acquired from blood or bodily fluids. Although many biomarkers have been used in clinical practices, there is still a need for biomarkers with more stability, sensitivity, and specificity.

CircRNAs occupy up to 1% of the total RNA and are differently (via specific isoforms) or specifically expressed in various types of cells (*Salzman et al., 2013*), suggesting that their expression could be associated with different conditions and consequently could serve as specific biomarkers. Furthermore, circRNAs are relatively abundant in different cells. Reports show that they show higher abundance in low-proliferating cells, such as in the brain, but are relatively lower in number in high-proliferating cells such as in the liver. It is hypothesized that as cells proliferate, circRNAs get divided into the daughter cells, resulting in a total lower abundance of the circRNAs in highly proliferating cells (*Bachmayr-Heyda et al., 2015*; *Nair et al., 2016*). This is also evident in highly proliferating tumor cells, which show lower levels of circRNAs than normal tissues (*Bachmayr-Heyda et al., 2015*).

*Memczak et al. (2015)* proved the existence of circRNAs in the blood. However, it is more important to know if the amount of circRNA in blood can be easily detected. Predictably, the abundance of circRNAs in blood is higher than that in tissues but lower than that in the brain. Considering that the low amounts of circRNAs in liver have been suggested as biomarkers (*Zhang et al., 2018b*), we propose that the relatively more abundant blood circRNAs may efficiently serve as biomarkers. Recent research confirmed that circRNAs are also abundant in exosomes, which can be a potential new way to employ circRNAs for disease diagnosis and prognosis (*Li et al., 2015b*). Although exosomes contain some unwanted proteins, DNA, and RNA, these molecules can act

as messenger shuttles between cells (*Pillay et al., 2017*). Therefore, exosomes are potential biomarkers, which have been used for diagnosing multiple diseases as well as assessing responses to drug treatments. Although circRNAs are lower in high-proliferating cells, they are higher in exosomes in such cells than in normal exosomes. Additionally, it is possible that a larger number of exosomes are shed from particular sites in the higher proliferating cells than in normal cells. Together, this suggests that detecting exosomes in such tissues may be more promising to diagnose diseases.

Although the total amount of circRNAs is low in body fluids such as blood, saliva, and gastric fluid and other tissues, it is still higher than that of linear RNAs. Multiple reports have proposed that circRNAs as disease biomarkers are superior to the corresponding mRNAs and lncRNAs in terms of abundance, stability, and specificity. The diagnosis and prediction of stage characteristics of a disease are also better with circRNAs than with mRNAs and lncRNAs. Additionally, it is easier and more convenient to detect circRNAs in blood samples, saliva, and gastric juice. Research has identified 422 circRNAs in human cell-free saliva, and these have been implicated in intercellular signaling and inflammatory responses (*Bahn et al., 2015*).

Cardiovascular disease has been reported to be responsible for 31% of all global deaths. Cardiovascular diseases include coronary artery disease (CAD), aneurysm, MI, and related diseases such as HF and pulmonary arterial hypertension (PAH). Traditional treatment strategies such as controlling risk factors and early treatment after diagnosis result in poor prognosis. Although various biomarkers for CVD have been used for several years, the diagnosis and subsequent treatment often occur very late as these biomarkers reach significant detectable levels at later disease stages. Brain natriuretic peptide has been used for the diagnosis of HF, cardiac TnT/I and CK-MB have been utilized for the diagnosis of MI, and D-dimer has been used for the diagnosis of aortic dissection. However, there is still a lack of biomarkers for CAD and other CVDs. In recent years, ncRNAs such as miRNAs and lncRNAs have been suggested as potential biomarkers for CVD. However, it is as yet unclear if circRNAs could be potential new biomarkers for CVD. Therefore, we discuss below recent discoveries that illustrate the potential of using circRNAs as diagnostic and prognostic CVD biomarkers.

## Coronary artery disease and atherosclerosis

Coronary artery disease has been a heavy burden on social economy mainly due to its late diagnosis and severe complications. Coronary artery disease and atherosclerosis are linked to endothelial injury and imbalance of lipid metabolism. The important roles of miRNAs in atherosclerosis, such as regulation of vascular smooth muscle cell function, lipid homeostasis, and cytokine responsiveness, have been described thoroughly (*Feinberg & Moore, 2016*; *Markus et al., 2016*). Myocardial infarction and HF are the main results of CAD and lead to severe mortality. CircANRIL can regulate cell functions related to atherosclerosis and plays an atheroprotective role. The antisense ANRIL and the INK4/ARF genes are found on chromosome 9p21.3 (*Wellcome Trust Case Control Consortium, 2007*). Patients with high circANRIL expression develop less severe CAD. This circRNA can impair the process of exonuclease-mediated pre-rRNA and

ribosome biogenesis by binding to PES1. Consequently, this induces apoptosis and inhibition of proliferation of vascular smooth muscle cells and macrophages, which can promote anti-atherogenic cell stability and prevent degradation (*Burd et al., 2010*). It also can affect atherosclerosis by regulating INK4/ARF expression. Thus, stable circANRIL may act as a potential biomarker and therapeutic target for atherosclerosis (*Sarkar et al., 2017*; *Holdt et al., 2016*).

*Pan et al. (2017)* used circRNA microarray to detect differentially expressed circRNAs between three samples from CAD patients and three control plasma samples. They found hsa_circ_0006323, hsa_circ_0032970, hsa_circ_0051172, hsa_circ_0054537, hsa_circ_0057576, hsa_circ_0068942, hsa_circ_0082824, hsa_circ_0083357, and hsa_circ_0089378 to be differentially expressed with fold change $\geq$ 1.5 and $P < 0.05$. These nine circRNAs act as an hsa-miR-130a-3p sponge to influence its target mRNA, transient receptor potential cation channel subfamily M member 3 (TRPM3). Transient receptor potential cation channel subfamily M member 3 regulates contractility and proliferation of vascular smooth muscle cells in coordination with cholesterol, which plays an important role in CAD. However, these results need to be verified using a larger sample size of CAD patients and normal individuals. *Zhao et al. (2017)* used the same method to show differences between the peripheral blood of 12 patients with CAD and 12 normal individuals and found that hsa_circ_0124644 and hsa_circ_0082081 were significantly associated with CAD. In addition, they found that hsa_circ_0124644 in blood could act as a potential diagnostic biomarker for CAD with a specificity and sensitivity of 0.626 and 0.861, respectively.

## Myocardial fibrosis

Myocardial fibrosis is a disease of myocardial stiffness, which can reduce myocardial shortening to induce diastole difficulty. Endothelial-to-mesenchymal transition (EndMT) is the process of inducing normal endothelial cells into mesenchymal-like cells. Tissue fibrosis is associated with fibroblast-specific protein-1 (FSP-1) expression and collagen deposition. The profibrotic factor, transforming growth factor β (TGF-β), is able to drive EndMT progression. It has been shown that TGF-β1-mediated induction of α-SMA expression and concomitant loss of VE-cadherin expression in aortic endothelial cells results in EndMT. Three EndMT-related circRNAs, chr5:90817794|90827570, chr8:71336875|71337745, and chr6:22033342|22038870, were found to be significantly upregulated in TGF-β1-treated rat coronary artery endothelial cells (CAEC). These three circRNAs may be potential biomarkers of EndMT-induced myocardial fibrosis (*Huang et al., 2018a*).

*Zhou & Yu (2017)* found 24 up-regulated circRNAs and 19 down-regulated circRNAs with fold change > 3 and $P < 0.05$ between diabetic db/db mice and db/m control mice. They selected one of the markedly increased circRNAs (circRNA_010567) for further bioinformatics analysis and found it to contain binding sites for miR-141, which is predicted to regulate TGF-β1 function. Further, results suggested that circRNA_010567 could directly target miR-141 and regulate TGF-β1 expression, which can mediate resection-associated fibrosis. These results imply that circRNA_010567 may play a key

role in myocardial fibrosis, and thus provide a novel insight into cardiopathy pathogenesis and act as a potential diagnostic biomarker.

*Tang et al. (2017)* reported a new target, circRNA_000203, which was significantly upregulated in diabetic myocardium. This circRNA can act as a sponge for miR-26b-5p to restrain its downstream targets, Col1a2 and CTGF, thus promoting pro-fibrosis effects. Although circRNA_000203 may play important roles in myocardial fibrosis, its efficacy as a biomarker needs further investigation.

## Cardiomyopathy

Cardiomyopathy is a disease with primary abnormalities in the structure and function of the heart. Dilated cardiomyopathy is a morphological subtype of this disease. *Siede et al. (2017)* reported increased expression of three circRNAs SLC8A1, CHD7, and ATXN10 relative to their host gene expression and decreased circDNA6JC expression in patients with dilated cardiomyopathy. *Xie et al. (2016)* found that Foxo3 is the protective agent in *Ganoderma* spore oil against cardiomyopathy. In addition, circAmotl1 was associated with cardiac dysfunction, wherein it could promote cardiac repair through binding AKT1 and PDK1 to accelerate the function of cardioprotective nuclear translocation of pAKT. This may provide a way to detect cardiac dysfunction before HF. circAmotl1 was also shown to be important for ameliorating the negative effects of Dox on the heart, such as resistant fibrosis, apoptosis, and hypertrophy. Thus, circAmotl1 could have therapeutic potential in the treatment of cardiomyopathy (*Zeng et al., 2017*). *Khan et al. (2016)* reported that RBM20-dependent TTN circRNAs play important roles in dilated cardiomyopathy. However, there are only few studies on the role of circRNAs in cardiomyopathy.

## Heart failure

Heart failure is a disease that is often a secondary effect caused by another disease. It is also a final outcome of some CVDs such as CAD, vascular disease, and MI. Therefore, finding an ideal biomarker for HF is challenging. Heart failure leads to a decrease in systolic and/or diastolic function (*Mentz & O'Connor, 2016*). Traditional standard biochemical biomarkers such as BNP have been used to distinguish HF from other conditions (*O'Donoghue et al., 2005*). However, studies have shown that ncRNAs are more sensitive than traditional biomarkers such as BNP or cardiac troponins. Up until now, more than 50 miRNAs and three lncRNAs have been described as biomarkers of HF.

*Salgado-Somoza et al. (2017)* reported use of the circRNA, myocardial infarction-associated circular RNA (MICRA), to predict the risk in MI patients. Using blood samples from 472 acute MI patients, they found that the expression levels of MICRA were lower in patients with ejection fraction (EF) $\leq 40\%$ than in patients with EF $> 41\%$. Patients with lower levels of MICRA were also at high risk of decreased EF (0.78 [0.64–0.95]). They further identified that MICRA was present in 86% of the samples, and bootstrap internal validation indicated that it could act as an optimal HF predictive biomarker, similar to the traditional markers (Nt-proBNP, creatine phosphokinase, CPK). *Wang, Chen & Sen (2016)* reported that a special heart-related circRNA (HRCR) may play an important role in HF by acting as an miR-223 sponge to increase the expression of apoptosis repressor with CARD domain (ARC); however, this requires further investigation.

## Myocardial injury

Myocardial injury and apoptosis are associated with HF and MI as well as reperfusion injury. Increase in apoptosis augments MI and HF, whereas reduced apoptosis protects the heart. *Wang et al. (2017a)* found that the circRNA, MFACR, could increase mitochondrial fission and apoptosis as well as cardiomyocyte cell death through downregulation of the ceRNA (competing endogenous RNAs), miR-652-3p, and promotion of MTP18 translation. In this study, MFACR was shown to act as a sponge to inhibit the activity of miR-652-3p, which directly targets MTP18 to inhibit mitochondrial fission and apoptosis. Thus, MFACR may act as a biomarker to predict apoptosis in the heart and may serve as a potential therapeutic target for treatment.

## Myocardial infarction

MI is a catastrophic condition caused by ischemic heart disease and can lead to heart tissue damage and mortality. Although revascularization can be used as a valuable method to stabilize the crisis, selective biomarkers are required to assess the risk and therapeutic response after the infarction (*Hachey et al., 2017*). Acute MI poses sizeable morbidity and mortality risks. Hsa-miR-122-5p levels that are different in the plasma of CVD patients were proposed as an early prognostic biomarker of acute MI (*Cortez-Dias et al., 2016*).

Through circRNA microarray analysis in mouse hearts with autophagy, *Zhou et al. (2018)* found that mmu_circRNA_006636 (ACR) was significantly decreased after I/R injury. ACR was found to regulate Pink1 expression through inhibition of DNA methylation of Pink1 by binding to DNMT3B. Further, after verifying the functional role of Pink1 in autophagy, they identified the downstream target of Pink1 as FAM65B. Thus, ACR can repress autophagy and MI by targeting Pink1-mediated phosphorylation of FAM65B, and could act as a potential therapeutic target and biomarker for ischemia/ reperfusion and MI.

*Geng et al. (2016)* found that circRNA CDR1AS could act as an miR-7a sponge to regulate the expression of its target gene (PARP and SP1) and interfere with its protective role in MI injury. In addition, apoptosis-related circRNA MFACR could regulate miR-652-3p to promote the progress of MTP18, which is relevant for MI (*Wang et al., 2017a*). *Vausort et al. (2016)* reported that MICRA was highly expressed in peripheral blood of MI patients. However, its potential value act as a biomarker for MI remains to be further investigated. *Deng et al. (2016)* found that circRNA_081881 was differentially expressed in plasma samples of acute MI patients. This report shows that the down-regulated circRNA_081881 can sequester miR-548 through competitive binding sites to reduce PPAR$\gamma$ (a heart-protective factor) expression, which is reduced in the plasma of patients with acute MI.

## Hypertension

Hypertension has become a major risk factor for the development of CVD and is highly prevalent across the world, even in young people. It has been a major contributor to morbidity and mortality and is considered as a socioeconomic burden. The diagnosis and treatment are influenced by many factors and there is a lack of useful biomolecules for clinical prediction and diagnosis.

*Bao et al. (2018)* detected differences in the expression of some circRNAs between five pairs of newly diagnosed essential hypertension (EH) and non-EH whole blood samples. They found that hsa_circ_0037911 levels in the EH group were significantly higher than those in the control group ($t = 2.834$, $P = 0.005$). In addition, they proved that the target hsa_circ_0037911 was an effective predictor of EH by using the receiver operating characteristic curve to investigate its diagnostic value (area under curve, AUC = 0.627; $P = 0.002$). Their study also showed that there was a direct correlation between hsa_circ_0037911 and serum creatinine, indicating that this RNA plays an important role in the pathogenesis of EH. Another study reported that 46 circRNAs were significantly upregulated and 13 downregulated (FC > 2.0 and $P < 0.05$) between human hypertensive plasma and normal plasma. hsa-circ-0005870 was verified to be significantly downregulated in hypertensive patients. Bioinformatics analysis also indicated that hsa-circ-0005870 may represent a novel biomarker for the diagnosis of hypertension (*Wu et al., 2017*).

## Aneurysm disease

Aneurysm disease especially aortic dissection is a life-threatening condition with a lethality rate of 1–2% per hour after onset of symptoms in untreated patients. The treatment is complex and the prognosis is poor, especially Stanford A aortic dissection. However, there is no available biomarker that can reveal this disease before acute onset. Therefore, predictive diagnosis and treatment are vital to improve the survival rate and to prevent severe complications.

*Zou et al. (2017)* reported 106 down-regulated and 156 up-regulated circRNAs between three normal patients and three thoracic aortic dissection (TAD) patients. They found hsa_circRNA_101238 to be upregulated with a fold change $\geq 1.5$, $P < 0.05$, and co-expression network revealed that only this target circRNA interacted with the three altered miRNAs (hsa-miR-320a, hsa-miR-320b, and hsa-miR-320c). The circRNA-miRNA-mRNA network predicted miRNA targets of hsa_circRNA_101238 were hsa-miR-320b, hsa-miR-320a, hsa-miR-138-5p, hsa-miR-593-5p, and hsa-miR-320c. Through target gene prediction and luciferase assays, they found that hsa_circRNA_101238 acted as an miR-320a sponge through inhibiting the expression of hsa-miR-320a to increase MMP9 expression, which may be involved in the pathogenesis of TAD. Thus, hsa_circRNA_101238 may be a potential biomarker for TAD. However, the function and mechanism of circRNAs in TAD need further experimental evidence.

## Pathological hypertrophy

Cardiomyopathies are myocardial diseases with morphological and functional abnormalities and can be classified as primary or intrinsic cardiomyopathies. Pathological diagnosis of cardiomyopathies requires invasive and potentially dangerous tests. Therefore, many patients with cardiomyopathies usually opt for biochemical tests that can help diagnosis (*Coats et al., 2015*).

*Meng et al. (2019)* reported differentially expressed circRNAs in cardiac hypertrophy cells cultured in the presence of high and normal levels of D-glucose. Five circRNAs,

namely ciRNA261, ciRNA26, circRNA1191, circRNA4251, and circRNA6913, were found to be significantly differentially expressed ($P < 0.05$ and fold change > 2 or < 0.5) and had more than 60 target miRNAs, which implied that these circRNAs may play important roles in cardiac hypertrophy and potentially serve as biomarkers.

*Wang et al. (2016)* reported that HRCR could act as an endogenous miR-223 sponge and inhibit the activity of miR-223. In addition, they also reported that miR-223 is functionally related to hypertrophy through ARC. Thus, HRCR inhibits miR-223 activity, resulting in increased expression of its downstream target ARC and inhibition of cardiac hypertrophy and HF in mice. Thus, HRCR may be a useful biomarker for diagnosis and prognosis of cardiac hypertrophy and HF.

### Cardiac senescence

Cardiac senescence accompanies aging and may decrease heart function. *Chen et al. (2018)* reported that aging significantly affects the cardiac muscle. By using high throughput RNA-Seq, they found 21 up-regulated circRNAs and one down-regulated gene in cardiac muscle during aging. Within the network, circRNA005698 was found to be associated with seven miRNAs. Further, they found that circRNAs including circRNA005698 might play a key role in regulating pro-coagulation process during aging. Thus, circRNA005698 can act as a biomarker for cardiac senescence.

*Du et al. (2017)* also found that circ-Foxo3 is relevant for cell senescence in doxorubicin-induced mouse cardiomyopathy. This circRNA could interact with senescence-related proteins (ID1 and E2F1) and stress-related proteins (HIF1α and FAK) in the cytoplasm, leading to the inhibition of the anti-stress and anti-senescent roles of these proteins and consequently promoting cardiac senescence. circFoxo3 is not only found in aged heart of mice but also in humans.

## GENERAL COMMENTS

### Conclusion

In summary, circRNAs are a class of ncRNAs that mainly regulate gene expression. Increasing evidence indicates that circRNAs are abundantly found in saliva, exosomes, and clinical standard blood samples, which make them promising biomarkers for disease diagnosis and outcome prediction. They are more stable and sensitive as well as specific than standard biomarkers. In CVD, secreted lncRNAs have been described as biomarkers for several conditions including MI, cardiac failure, and atrial fibrillation. Hopefully, in the future, the use of circRNAs as biomarkers will become routine in clinical practice.

### Limitations

To be able to use circRNAs as routine biomarkers, there are some limitations that need to be addressed. First, several methodological factors including sample collection and processing, as well as assay performance and ncRNA quantification can influence the quality of the resulting data and need to be improved. Second, analysis of case studies can be limited by the sample size due to lack of statistical power. In addition, sex, age, and the diversity of the cardiovascular risk factors may introduce bias in limited samples.

Therefore, large cohorts and multi-center studies are necessary to interpret data and make conclusions. Third, patients with CVD are likely to take anticoagulant drugs such as aspirin and clopidogrel before blood collection, which can alter circulating ncRNA concentrations and thereby their quantification. Fourth, primer design and normalization in quantitative real-time Pol chain reaction can influence the outcome of biomarker studies. Therefore, it is necessary to standardize such procedures to eliminate technical and analytical variability.

## ABBREVIATIONS

| | |
|---|---|
| **ncRNAs** | noncoding RNAs |
| **miRNAs** | microRNAs |
| **snRNAs** | small nuclear RNA |
| **piRNAs** | piwi-interacting RNA |
| **siRNAs** | small interfering RNA |
| **lncRNAs** | long noncoding RNAs |
| **CVD** | cardiovascular disease |
| **circRNAs** | circular RNAs |
| **MACE** | major adverse cardiovascular events |
| **MI** | myocardial infarction |
| **BNP** | beta-natriuretic peptide |
| **HF** | heart failure |
| **RNA-Seq** | RNA sequencing |
| **RBP** | RNA-binding proteins |
| **QKI** | Quaking |
| **MBL** | Muscleblind |
| **ADAR1** | adenosine deaminases acting on RNAs |
| **hnRNP** | heterogeneous nuclear ribonucleoprotein |
| **SR** | serine-arginine |
| **FMRP** | fragile mental retardation protein |
| **AGO2** | Argonaute 2 |
| **PABPN1** | polyadenylate binding protein 1 |
| **HuR** | human antigen R |
| **ANRIL** | antisense non-coding RNA in the INK4 locus |
| **ANKRD52** | ankyrin repeat domain 52 |
| **Pol** | polymerase |
| **PES** | Pescadillo |
| **IRES** | internal ribosome entry site |
| **RCA** | rolling circle amplification |
| **HDAg** | hepatitis D virus antigen |
| **CAD** | coronary artery disease |
| **PAH** | pulmonary arterial hypertension |
| **TnT/I** | troponin T/I |

| TRPM3 | transient receptor potential cation channel subfamily M member 3 |
| EndMT | Endothelial-to-mesenchymal transition |
| FSP-1 | fibroblast-specific protein-1 |
| TGF-β | transforming growth factor β |
| CAEC | coronary artery endothelial cells |
| CPK | creatine phosphokinase |
| HRCR | heart-related circRNA |
| ARC | apoptosis repressor with CARD domain |
| MICRA | myocardial infarction-associated circular RNA |
| EH | essential hypertension |
| TAD | thoracic aortic dissection. |

### Funding
The authors received no funding for this work.

### Competing Interests
The authors declare that they have no competing interests.

### Author Contributions
- Weitie Wang conceived and designed the experiments, performed the experiments, analyzed the data, contributed reagents/materials/analysis tools, prepared figures and/or tables, approved the final draft.
- Yong Wang conceived and designed the experiments, performed the experiments, contributed reagents/materials/analysis tools.
- Hulin Piao conceived and designed the experiments, analyzed the data, contributed reagents/materials/analysis tools.
- Bo Li conceived and designed the experiments, contributed reagents/materials/analysis tools.
- Maoxun Huang conceived and designed the experiments, performed the experiments, contributed reagents/materials/analysis tools.
- Zhicheng Zhu conceived and designed the experiments, performed the experiments, analyzed the data, contributed reagents/materials/analysis tools.
- Dan Li conceived and designed the experiments, analyzed the data, contributed reagents/materials/analysis tools.
- Tiance Wang conceived and designed the experiments, contributed reagents/materials/analysis tools.
- Rihao Xu conceived and designed the experiments, contributed reagents/materials/analysis tools.
- Kexiang Liu conceived and designed the experiments, performed the experiments, analyzed the data, contributed reagents/materials/analysis tools, prepared figures and/or tables, authored or reviewed drafts of the paper, approved the final draft.

## Data Availability

This is a review manuscript; no raw data was generated.

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
