# Peer review of "Circular RNAs as potential biomarkers and therapeutics for cardiovascular disease"

_PeerJ, doi:10.7717/peerj.6831_

## Round 0.1 · original submission · Major Revisions

Both reviewers had critical remarks, please take it into account. The second reviewer provided marked up questions in the PDF file attached.

I have concern regarding a similar recent publication -
J Cell Physiol. 2018 Oct 20. doi: 10.1002/jcp.27384. [Epub ahead of print] Circular RNA in cardiovascular disease

It looks very similar. Please check it so that you are presenting original manuscript, maybe even changing the paper title.

Despite the critical remarks I encourage you resubmit your updated manuscript to PeerJ

·

Basic reporting

English renders proofreading

Experimental design

N/A

Validity of the findings

N/A

Additional comments

Review on the article titled “Circular RNAs as potential biomarkers and therapeutic for cardiovascular disease” by Wang et al., 2019
The authors present a comprehensive review encompassing implication of circRNA in cardiovascular disease (CVD) factors. The review presents the mechanistic insights of circRNA biogenesis, genetic aspects of CVD factors and a review of current research on implication of circRNA on CVD.
While the structure and style of review is acceptable, the numerous grammar problems renders English proofreading. Also, biogenesis of circRNA description is currently non-transparent and should be reworked according to the notes below.
Notes:

1. P. 6 (43-44) “In human genome, noncoding RNAs (ncRNAs) account for the majority of transcripts and only 1~2% are protein-coding genes” – please reference this statement. Protein coding genes comprise approximately 40% of the genome. The transcripts usually correspond to full-length genes. If this is even true in terms of numbers/percentage, it should be referenced correctly, not comparing transcripts to genes (“…transcribed from protein coding genes”).
2. P6( 44) housekeeper ncRNA – housekeeping RNA? Never heard of these, please reference/explain.
3. First paragraph of the intro should present circRNA as well, while currently doesn’t.
4. The approximate numbers/percentage of each RNA class should be presented in human genome within the abovementioned paragraph.
5. P6 (63) “since long”? <=>“for long time?”
6. P8(98): liner = linear
7. The first paragraph of circRNA biogenesis chapter (98-109) should extensively reference Fig. 1 with corresponding indices (a-d) for each instance, in contrast of referencing the figure at the end of the paragraph. This paragraph is currently not transparent and is hard to apprehend. It should be extended for more clearly manifested/illustrated details and with the intense referencing to the figure, since it’s a vital chapter for further reading. It can be done by using Fig. 1 as a directory to consequently describe each variant, extensively explaining (not deciphered in Fig. 1. legend) terms like ecircRNA, EIciRNA.
8. P9 (112-113): “can stop (prevent?) circRNA formation by melting the stem structure” Relating to above: where is stem structure in the figure? Is it fig 1 b? It should be referenced to the figure in this instance.
9. P11(186): “is exist”
10. Fig 1 should preferentially be arranged for a vertically oriented page, with 4 cases presented separately per string(s). The typeset should be larger. All abbreviations in legends should be presented therein. Fig. 2 has some of denotations placed on the dark red/violet background that are not readable, typeset should be also made bigger, at least in legend.

Reviewer 2 ·

Basic reporting

The article discusses recent reports on the assessment of the biological significance of circular non-code RNAs and their possible diagnostic and prognostic value as markers of cardiovascular diseases. This topic is very relevant. In recent years, numerous experimental and review articles have been published, discussing the role of circular non-coding RNA in the regulation of transcription and translation processes in cells. Several review articles have been published, discussing the possibility of using circular non-coding RNAs as markers.
The article is written in good English. Article written in professional English.
The review is of broad and interdisciplinary interest that meets the requirements of the journal.

Experimental design

The research methodology corresponds to a comprehensive, unbiased coverage of the subject.
Literature is cited adequately.
The text of the review is well structured and organized logically in the form of coherent subsections.

Validity of the findings

The text is well thought out and consistent with the objectives for the purposes set out in the introduction.
In addition to the Conclusion, the article presents the Limitations section, where the factors limiting the use of non-coding circular RNA as biomarkers for diagnosis and diseases outcome prediction.

Additional comments

Сomments on the text of the article are given in the attached file.

Annotated reviews are not available for download in order to protect the identity of reviewers who chose to remain anonymous.

---

## Round 0.2 · Minor Revisions

There are still some remarks from second reviewer demanding minor revision. Please consider the annotated manuscript available through the link in this email. I believe you can update the manuscript fast.

·

Basic reporting

The authors addressed issues of circular RNA research, which definitely is a novel and important research area. It is relevant not only to cardiovascular disease area, but to cancer research, and many others.

Experimental design

Acceptible

Validity of the findings

N/A

Additional comments

As of now I have no major concerns to the manuscript.

Reviewer 2 ·

Basic reporting

The review is of broad and interdisciplinary interest that meets the requirements of the journal.

Experimental design

The research methodology corresponds to a comprehensive, unbiased coverage of the subject.
Literature is cited adequately.
The text of the review is well structured and organized logically in the form of coherent subsections.

Validity of the findings

The text is well thought out and consistent with the objectives for the purposes set out in the introduction.

Additional comments

A few more comments on the text of the article are given in the attached file.

Annotated reviews are not available for download in order to protect the identity of reviewers who chose to remain anonymous.

---

## Round 0.3 · accepted · Accept

Thank you for the update. Since previous version had only minor remarks I think it could be directly accepted now without additional review round. Circular RNA overall is important topic to be discussed in the publication.

#